# Do Symbolic or Black-Box Representations Generalise Better In Learned Optimisation?

## Abstract

Until recently, behind *every* algorithmic advance in machine learning was a human researcher. Now, however, algorithms can be *meta-learned automatically*, with little human input. However, to be truly useful, such algorithms must generalise beyond their training distribution. This is especially challenging in reinforcement learning (RL), where transferring algorithms between environments with vastly different dynamics is difficult and training on diverse environments often requires prohibitively expensive large-scale data collection. Learned optimisation is a branch of algorithmic discovery that meta-learns optimiser update rules. Learned optimisers can be classified into two groups: black-box algorithms, where the optimiser is a neural network; or symbolic algorithms, where the optimiser is represented using mathematical functions or code. While some claim that symbolic algorithms generalise better than black-box ones (Chen et al., 2023), testing such assertions is complicated by the fact that symbolic algorithms typically include additional hyperparameters, and thus their evaluation is done *many-shot*. This is an unfair comparison with the *zero-shot* evaluation of black-box optimisers. In this work, we build a pipeline to discover symbolic optimisers which are *hyperparameter-free*, enabling a fair comparison of the generalisation of symbolic optimisers with that of an open-source state-of-the-art black-box optimiser trained for RL[1]. Based on our analysis, we propose suggestions to improve the symbolic optimiser discovery pipeline for RL, with an overall objective of reducing the need for hyperparameter tuning to train an agent.

## 1 Introduction

Improvements to optimisation algorithms have driven machine learning to new heights over the past few decades. The introduction of components like gradient momentum, second order momentum (Nesterov, 1983; Kingma & Ba, 2017) and adaptive learning rates (Kingma & Ba, 2017; Zhuang et al., 2020) has enabled swifter and more stable convergence, while learning rate annealing has improved the fidelity of converged solutions. Recent evidence (Andrychowicz et al., 2016; Chen et al., 2021; Metz et al., 2022c; Chen et al., 2023; Goldie et al., 2024) suggests that the improvement of optimisers could be automated via *learned optimisation*. In learned optimisation, developing new optimisation algorithms is itself a *meta*-learning process based on data.

Approaches to learned optimisation fall into two camps. Most work (e.g., (Metz et al., 2022a; Kirsch & Schmidhuber, 2022; Andrychowicz et al., 2016; Wichrowska et al., 2017; Goldie et al., 2024)) replaces the optimiser, such as Adam (Kingma & Ba, 2017), with a black-box function using a neural network. In this scenario, the weights of the network are updated in an *outer loop* to maximise the performance of a trained model at the end of an *inner loop*. By contrast, some recent work (Chen et al., 2023; Song et al., 2024a) focuses on discovering *symbolic* optimisation algorithms. In this case, the optimiser is represented by a set of mathematical equations or programming instructions. In general, interest in symbolic algorithm discovery has grown in the past couple of years (Romera-Paredes et al., 2024; Lu et al., 2024a) due to the advent of large language models (OpenAI et al., 2024; Dubey et al., 2024, LLMs). There are arguments in favour of both approaches: black-box algorithms *may* be easier to work with (Goldie et al., 2024), while symbolic optimisers *may* generalise better (Chen et al., 2023). However, there exists little study into the veracity of these claims.

---

[1]Code to be released upon acceptance.

Furthermore, direct comparison between the approaches is complicated by the fact that they target subtly different problems; black-box optimisers are typically evaluated zero-shot without any tuneable hyperparameters, whereas symbolic optimisers such as Lion (Chen et al., 2023) tune hyperparameters *per-task*, making evaluation *many-shot*. Therefore, it is hard to compare these different paradigms like-for-like based on current literature.

The need for general optimisation algorithms is exacerbated in reinforcement learning (Sutton & Barto, 2018, RL) due to its many idiosyncratic issues which make optimisation challenging. In particular, RL is very sensitive to hyperparameters (Eimer et al., 2023) which can cause catastrophic instability if they are not correctly tuned. This instability may stem from the fact that RL often uses algorithms imported from supervised learning, motivating the development of RL-specific approaches (Henderson et al., 2018; Sarigül & Avci, 2017). For instance, many conventional optimisers, like Adam Kingma & Ba (2017), are designed for stationary learning tasks and are thus ill-suited for the non-stationarity of RL (Igl et al., 2021; Bengio et al., 2021). Learned optimisers tailored for RL show promise in addressing these issues (Lan et al., 2024; Goldie et al., 2024).

However, simply relying on a large meta-task diversity to enable generalisation across RL is impractical. For anything beyond simple environments, sampling in RL is expensive. Therefore, finding learned optimisation strategies which demonstrate generalisation, whilst maintaining a limited meta-training cost, would significantly improve the practicality of RL. In this work, we compare the generalisation capabilities of a pretrained, black-box optimiser for RL (Goldie et al., 2024) with a roughly equivalent symbolic optimiser discovered using an evolutionary process based around LLMs. We focus on a regime in which optimisers can only be learned from a small number of environments; we believe this represents a scenario of greater interest than training in a distribution of gridworlds, which has been a previous focus for generalisation (Goldie et al., 2024; Lan et al., 2024) but does not transfer well to the modern LLM-driven discovery pipeline. In doing so, we explore the question of whether black-box or symbolic optimisers are *actually* best for generalisation across a number of axes, including to different environments and to longer training lengths. We use these findings to recommend promising directions for future work in this field, thus providing a pathway to unlock truly general learned optimisation algorithms.

## 2 BACKGROUND

**Optimisation**    Optimisation is ubiquitous throughout machine learning. Given a general training objective $f_\theta(\cdot)$, there is an extensive set of optimisation algorithms whose goal is to guide $\theta$, a model's parameters, to the optimal $\theta^*$. Most fundamental of optimisers is gradient descent, where $\theta$ is updated iteratively towards negative gradient as $\theta_{t+1} \leftarrow \theta_t - \eta \nabla_\theta f(\cdot)$, using a step-size $\eta$.

A number of augmentations are frequently applied to gradient descent to enable quicker convergence, less noisy updates or improved asymptotic performance. For instance, modern optimisers like Adam (Kingma & Ba, 2017) and RMSProp (Tieleman et al., 2012) use *momentum*, a time-based moving average of gradients or updates which provides more consistent updates over training. Similarly, learning rate *annealing* or *warmup* change the step size over time to provide closer convergence to the optimum by the end of training, or improved stability at the beginning of training, respectively (Robbins, 1951; Gotmare et al., 2018).

**Reinforcement Learning**    Reinforcement learning focuses on Markov Decision Processes (Sutton & Barto, 2018, MDPs), defined as $\langle \mathcal{A}, \mathcal{S}, T, R, \rho, \gamma \rangle$. The *agent* learns a policy $\pi(\cdot|s_t) \in \Pi$ and, at each discrete timestep $t$, samples an action $a_t \in \mathcal{A}$ based on the current state $s_t \in \mathcal{S}$ (where $s_0 \sim \rho$). After sampling an action, the agent transitions to the next state $s_{t+1} \in \mathcal{S}$ according to a transition distribution $T(s_{t+1}|s_t, a_t)$ and receives a reward according to the reward function $R(s_t, a_t)$. The policy is trained to maximise the *discounted expected return*, $J^\pi$, based on the discount factor $\gamma \in [0, 1)$, which is defined over a fixed length episode as

$$J^\pi := \mathbb{E}_{a_{0:\infty} \sim \pi, s_0 \sim \rho, s_{1:\infty} \sim T} \left[ \sum_{t=0}^{T} \gamma^t R_t \right]. \tag{1}$$

Sample complexity is a major issue in reinforcement learning. Due to the potential cost of interacting with the environment, it can often be prohibitively expensive to collect large datasets. One opportunity to reduce sample complexity is to remove the reliance on hyperparameters intrinsic to RL. Learned optimisers without hyperparameters could help to unlock this capability.

**Optimisation Difficulties in RL**  Goldie et al. (2024) discuss three optimisation difficulties present in RL: plasticity loss (Lyle et al., 2023; 2022), a phenomenon in which neural networks *lose* the ability to learn when given new data; exploration, where the optimiser must escape local optima from the agent being trapped in a localised state-action space; and non-stationarity (Igl et al., 2021), which arises as the input and output distributions in RL are continuously changing. OPEN incorporates a number of features to tackle each individual problem. To be specific:

- For plasticity, OPEN conditions on neuron dormancy (Sokar et al., 2023), a metric which measures what proportion of a layer's activation comes from a specific neuron. Near-zero dormancy neurons are dormant and need to be reactivated. OPEN also learns separate update rules for each layer by conditioning on *layer proportion*.

- For nonstationarity, OPEN conditions on two timescales: *batch proportion*, or progress through epochs with the current batch of data; and *training proportion* (Jackson et al., 2023a), meaning how far through the training horizon optimisation is.

- To boost exploration, OPEN introduces stochasticity of a learned variance to the update. This enables similar exploration behaviour to parameter space noise (Plappert et al., 2018) or noisy nets (Fortunato et al., 2019) while also incidentally helping with dormancy.

## 3 RELATED WORK

**Meta-Learning Algorithms**  Meta-learning intends to replace handcrafted algorithms with ones learned from data. Though some approaches use *meta-gradients* which are backpropagated through training episodes (e.g., (Lan et al., 2024; Oh et al., 2020)), this is impractical in our setting. Firstly, meta-learning in RL requires long horizon rollouts, where untruncated backpropagation experiences exploding or vanishing gradients but truncating biases towards greedy algorithms (Wu et al., 2018; Metz et al., 2022b; Lu et al., 2022b). Secondly, with a *symbolic* optimiser, it is not obvious how to project gradients on to the non-numerical symbols of our algorithm, requiring more complex techniques (e.g. (Kuang et al., 2024; Chen et al., 2024)).

Evolutionary methods (Rechenberg, 1973; De Jong, 2006) provide an alternative. These are derivative-free optimisation methods which mutate and evaluate a populations of candidates. Common evolutionary methods include genetic algorithms (Such et al., 2018), covariance matrix adaptation (Hansen & Ostermeier, 2001), evolution strategies (Salimans et al., 2017) or, in the symbolic case, genetic programming (Koza, 1992). Evolution involves sequentially sampling population members, randomly changing their parameters and evaluating the final performance of the candidate. By optimising based on the final evaluation, rather than backpropagating *through* the rollout, evolutionary methods avoid many of the issues with meta-gradients.

Since the advent of LLMs, a new form of symbolic evolution has emerged (Romera-Paredes et al., 2024). Rather than applying *random* mutations, recent methods have replaced the evolutionary system with LLMs that suggest edits and reason about performance to guide search (Lu et al., 2024a; Meyerson et al., 2024; Lehman et al., 2022; Shojaee et al., 2024). This uses an LLM's prior knowledge to make 'intelligent' changes, in effect limiting the search to reasonable if not limited edits. Despite its recent invention, this technique has led to impressive results in function discovery (Romera-Paredes et al., 2024) or solving symbolic regression tasks (Shojaee et al., 2024).

**Learned Optimisation**  Learning to optimise (Metz et al., 2020; 2022c;a; Chen et al., 2023; Goldie et al., 2024, L2O) automates the discovery of better optimisers by *meta-learning* the algorithms. Generally, L2O replaces the optimiser with a neural network which conditions on the gradient, and potentially extra features, and outputs an update for *each parameter* in the training model. This method has proven effective in supervised and unsupervised learning (Metz et al., 2022c), but naïvely fails to transfer to RL. Due to the opportunity of learning *specialised* optimisation algorithms, OPEN (Goldie et al., 2024) and Optim4RL (Lan et al., 2024) L2O directly for RL. This is justified by many works suggesting RL-specific algorithms are warranted (Henderson et al., 2018; Bengio et al., 2021; Sarigül & Avci, 2017). Whereas Optim4RL attempts to L2O in RL by constraining the structure of the update, OPEN targets a number of difficulties present only in RL. Unfortunately, while these works have demonstrated signs of life for generalisation, there is little work exploring whether black-box optimisation is the best route to discover truly generalist optimisers.

An alternative approach is Lion (Chen et al., 2023), an optimiser discovered by *symbolic evolution*. However, to enable comparison between black-box and symbolic optimisation, we make a number

of key design changes from Lion. Firstly, our method searches in a **code**, rather than mathematical, parameterisation. This enables a richer space of functions by allowing conditional statements, like $(\texttt{if}, >, <)$. Secondly, by building on modern LLM-based methods, we diverge from Lion's naïve mutation operation. Since we attempt to *directly* compare against OPEN, whose inputs expands the algorithm design space drastically, the prior knowledge of an LLMs limits search to grounded mutations, thus preventing an excessive computation budget. Finally, we direct our search towards hyperparameter-free optimisers for RL to enable a fair comparison with OPEN.

**LLM-Guided Research** LLMs have increasingly been used for evolution-like optimisation recently (Song et al., 2024b). FunSearch (Romera-Paredes et al., 2024) demonstrated the validity of this approach by prompting an LLM to write functions for specific tasks. Like FunSearch, many works have synthesised the expressiveness of code with the creativity of LLMs: Hu et al. (2024) use LLMs to *design* agents for complex problems; DiscoPOP (Lu et al., 2024a) finds new objectives for preference optimisation in LLMs; and Lehman et al. (2022) incorporate Quality-Diversity approaches (Mouret & Clune, 2015) to produce different robot morphologies. While a common thread exists between these works and ours – using LLMs as a mutation operator for evolution – our discovery pipeline differs in its end-goal of learning an *optimisation algorithm*. We also consider how an LLM can be used to handle additional inputs, defined by OPEN, with natural language descriptions. Finally, we are approaching this setting from a purely analytical perspective.

# 4 MOTIVATION

To motivate our study into the generalisation capabilities of symbolic and black-box optimisers, we briefly compare the two in terms of potential advantages, grounded in both literature and intuition.

**Black-Box Optimisers** Since black-box optimisers are principally neural networks, they have a number of inherent advantages. Firstly, since they typically use *small* networks, they can easily be trained with evolution (Salimans et al., 2017) to avoid issues of short-term bias from truncated meta-gradients (Wu et al., 2018; Lu et al., 2022b). This does, however, have the issue of high memory usage and training sample complexity since each meta-update needs a number of full training loops equal to the population size. Though GPU-vectorisation (Bradbury et al., 2018) helps speed up this training dramatically (Lu et al., 2022b), it can require both high-end hardware and easy-to-sample environments which may not be practical.

Also, the simplicity of introducing additional inputs to black-box optimisers was demonstrated by OPEN, as well as an ease to learn interactions between input variables. This ability to easily scale with inputs may make black-box optimisers the best option in some settings.

Finally, due to their iterative meta-learning process, black-box optimisers can converge Goldie et al. (2024). This is in contrast to symbolic optimisers, which may not converge due to the mechanisms of symbolic evolution. This convergence can have advantages – training is predictable and usually stable – but can also lead to the optimiser being trapped in subpar optima.

**Symbolic Optimisers** Though symbolic discovery of optimisers is relatively unexplored, it has a number of *potential* advantages. It is worth noting, however, that we focus on a novel evaluation regime which aligns symbolic and black-box optimisation. Whereas Lion (Chen et al., 2023) needed *tuning* for its hyperparameters, black-box optimisers are applied zero-shot to new environments. Therefore, we concern ourselves with symbolic algorithms which *do not use hyperparameters*.

In this paper, we assess how black-box and symbolic optimisation algorithms generalise. Chen et al. (2023) suggest, without justification, that symbolic algorithms *should* generalise better, which seems intuitive. Symbolic optimisers are usually simpler; whereas Lion is 8 lines of code, OPEN uses up to $\sim 4000$ parameters, increasing the opportunity for overfitting. Also, symbolic optimisers must start from *something*, meaning they can be initialised from pre-existing optimisers.

A key advantage of symbolic algorithm discovery is that LLMs can interface into the discovery pipeline to improve the search efficiency, leaning on their vast knowledge-base to find new algorithms Lu et al. (2024a); Romera-Paredes et al. (2024). This also gives a large amount of control to the human-in-the-loop. As a researcher can describe design specifications in natural language, the search can be biased towards algorithms based on design requirements. We find this can help with including additional inputs to the algorithms, such as those from OPEN, even if the inputs are not included in the LLM's training data.

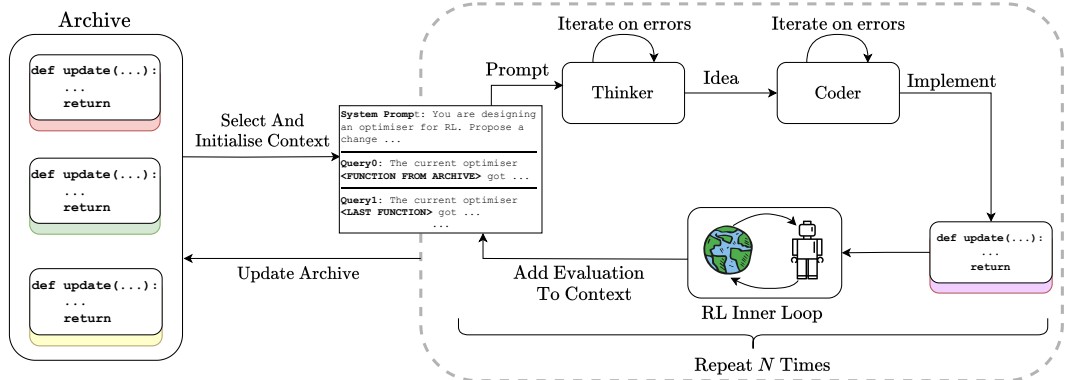

Figure 1: An overview of our discovery pipeline. An archive stores optimisers from previous generations. These are selected and used to initialise the LLM's context. A 'thinker' LLM proposes an idea which the 'coder' LLM interprets and implements, producing a new optimiser. The new optimiser is evaluated, added to the context for the thinker, and the process repeats for a finite number of steps before all optimisers are added to the archive and the outer loop progresses.

## 5  THE SYMBOLIC OPTIMISER DISCOVERY PIPELINE

We design a symbolic discovery loop to enable like-for-like comparison with OPEN which incorporates all of the features proposed in OPEN and described in section 2. We focus our comparison on the 'Multi-Task Training' setting from Goldie et al. (2024), where we meta-train on a small number of environments from MinAtar (Young & Tian, 2019; Lange, 2022). We believe this scenario is particularly interesting due to its correspondence with learning from a small number of fast proxy-tasks that approximate an ultimate objective.

We use an LLM in place of standard mutation in our system for the reasons mentioned in section 4. This lets us describe the inputs from OPEN in natural language to direct the search to 'reasonable' suggestions, avoiding a potentially more expensive and sample-inefficient random search, like Lion (Chen et al., 2023). However, LLMs can be notoriously fickle (Anagnostidis & Bulian, 2024; Gu et al., 2022). Therefore, we introduce a number of design decisions, described in this section, to improve the system's robustness. While we use GPT-4o (OpenAI et al., 2024) in this work, we believe that our system should also maintain robustness for weaker, open-source models (e.g. (Dubey et al., 2024; DeepSeek-AI et al., 2024)). We report discovery hyperparameters in Appendix A.

### 5.1  OVERVIEW

Figure 1 shows our discovery pipeline, which is similar to a number of 'LLM-Discovery' methods (Romera-Paredes et al., 2024; Lu et al., 2024a;b; Hu et al., 2024; Faldor et al., 2024), visually. At the start of the process, an archive is initialised with a set of candidate optimiser functions. After these are evaluated, one optimiser is selected for a generation of refinement, which involves iterative mutation by an LLM, followed by evaluation and insertion to the archive, for $N$ steps. After refinement is complete, a new optimiser is sampled and the process repeats.

Below, we introduce high-level design decisions which are detailed in the remainder of section 5.

**Initialising The Archive**   We follow DiscoPOP (Lu et al., 2024a) and Lion (Chen et al., 2023) by initialising training from a small set of optimisers. However, whereas DiscoPOP use pre-established loss functions, there is little precedent for hyperparameter-free optimisation. Therefore, we introduce a small number of hyperparameter-less optimisers by hand. These are designed to be flexible, while ensuring they don't fail catastrophically in the training environments.

**Selection**   Our pipeline periodically samples a new optimiser to refine at each generation, aligning closely to traditional evolutionary computation. This contrasts with, say, DiscoPOP (Lu et al., 2024a), which uses one long conversation with an LLM. By using the LLM more sparingly, this approach has the added benefit of potentially letting our system operate with less powerful language models. We select the best optimisers from the archive with probability $p$, and select random optimisers from the archive with an exploration probability $1 - p$.

**Mutation** We split mutation over two LLMs: a *thinker*, which proposes a new idea based on the current optimiser's performance; and a *coder*, which implements the proposed changes. This separation ensures faithful interpretations of ideas in the implementation and provides additional user control with the different prompts. Our thinker prompt also includes examples of performant optimisers in each environment.

**Evaluation** We evaluate optimisers on full-length RL environments at every refinement step. We track the final return and return area-under-the-curve of each optimiser for the thinker's context to enable in-context reasoning. To sidestep the problem of score aggregation over multiple environments faced by OPEN, we simply give the LLM returns for all environments and prompt it to maximise performance in all.

## 5.2 INITIALISATION

Similar to recent works (Lu et al., 2024a; Faldor et al., 2024; Hu et al., 2024; Chen et al., 2023), we initialise the archive of optimisers to a set of reasonable functions. However, given the scarcity of research on hyperparameter-free optimisation, the selection of initial optimisers is not straightforward. To address this, we create a few sensible optimisers to kickstart learning. In most cases, we write simple functions which have scaled *relative* changes to weights, though we also include a simple LLM-proposed function for diversity.

All optimisers follow the same design principles: they are simple, so that there are a large number of possible directions to improve them; they are diverse, so that they can lead to very different optimisers after refinement; and they are hyperparameter-free, meaning that any values are fixed for all environments. Notably, our initial optimisers only depend on the parameter value and the gradient, allowing the LLM to discover creative ways to use the additional inputs from OPEN without undue bias. We include all of the initial optimisers in appendix B.

## 5.3 EVOLUTION

For discovery, we blend LLM-based discovery algorithms with more conventional evolution (e.g. (Koza, 1992)). In doing so, we exploit the reasoning capabilities of LLMs to propose intelligent in-context changes while leveraging population-based evolution. The process runs as follows:

At the start of a new generation, we sample an 'initial' optimiser (section 5.3.1) and set of context optimisers (section 5.3.3) and prompt the LLMs to make small optimiser edits for a fixed number of *refinement steps*, $N$. At each refinement step, we evaluate the optimiser on *all* RL environments after a full RL inner-loop. Like OPEN, we use PPO Schulman et al. (2017) as the RL algorithm. After each generation, we add *all* evaluated optimisers to the archive and sample a new initialisation and context. Therefore, like Faldor et al. (2024), our archive grows over meta-training.

### 5.3.1 SAMPLING NEW OPTIMISERS

We sample a new 'base' optimiser each generation. To balance *exploration* and *exploitation* in our discovery process, we mostly sample *good* optimisers while occasionally selecting randomly to promote diversity. However, the notion of *good* or *bad* is not black and white when considering multiple environments of different reward scales. Naïvely averaging returns will prioritise environments which have a large reward scale, while normalising by, say, Adam's (Kingma & Ba, 2017) performance biases selection to environments where Adam underperforms (Goldie et al., 2024).

Instead, we use the average of per-environment rankings, based on return, over the population to measure how successful an algorithm is. In addition to scale-invariance, this has the benefit of weeding out optimisers which overfit to one environment, aiding robustness. After calculating the average rankings for the population, we select high-ranking optimisers with a probability $p$ and sample from the full population with probability $(1-p)$. In this work, we set $p = 0.8$ to balance sample efficiency (*mostly* starting from a performant optimiser) with diversity (occasionally sampling random optimisers).

### 5.3.2 MUTATION

We find that there is an occasional disparity between the proposal and implementation from LLMs when prompted naïvely. This hurts interpretability; it is not possible to tell what changes the LLM is making purely by observing the conversation. Therefore, we augment our system into a 2-LLM

setup by dividing out *thinking* and *coding*. The *thinker* has the responsibility of suggesting changes to the currently sampled optimiser and explaining why this change might be helpful. The *coder* has the task of converting the proposed idea into a code edit and implementing a syntactically correct, faithful python function. As an additional benefit, this allows different prompting strategies for each operation, giving the user additional control over the discovery trajectory.

### 5.3.3 PROMPTING

Different prompts can lead to vastly different results when using LLMs (Anagnostidis & Bulian, 2024; Gu et al., 2022). Here, we discuss the design decisions made in our prompting, and provide examples of the actual prompts in Appendix C.

**Difficulties in RL**    To enable intelligent suggestions based on the problems of RL from OPEN, described in section 2, we provide a high level overview of each additional input variable and what typical values might mean.

**Previous Performance**    To leverage in-context suggestion making, we condition the thinker on the returns of the current optimiser and randomly sampled 'context optimisers', which perform well in individual environments. To avoid issues highlighted in Goldie et al. (2024), where aggregating scores between different environments proved difficult, we include final return values for all environments into the prompt directly without averaging. This encourages the LLM itself to balance improvements between environments. To boost in-context reasoning further, we also provide values for the area-under-the-curve.

**Separating Prompts**    To ensure fulfilment of their separate roles, we prompt the thinker and coder LLMs differently. The thinker is prompted to produce a new idea based on previous performance while the coder converts the idea into a code update. Whereas the thinker is prompted with a *history* of optimisers for reasoning, the coder receives only the current optimiser and proposed change to avoid obfuscating its task. Separating thinking and code has been shown to improve performance in other work (Ye et al., 2024; Liu et al., 2024).

**Design Suggestions**    For both the coder and thinker, we propose a number of considerations to aid discovery. For instance, in the thinker we emphasise coming up with creative solutions, a need for generalisation and the necessity of not introducing new hyperparameters. For the coder, we focus on faithfulness and correctness, in addition to requesting commented code for interpretability.

## 6 DISCOVERY RESULTS

In figure 2, we show the meta-training curve for the symbolic discovery process. Notably, we find that, despite only selecting for high *average* fitnesses, our discovered symbolic optimisers have *consistently* high rankings across the four training environments. We also compute rankings for OPEN and Adam, with a standard *untuned* learning rate of $1e$-3. Based on their ranking compared to the population, neither Adam nor OPEN has robust performance across *all* environments. Below, we show the three highest average rank discovered optimisers which form the basis of our analysis.

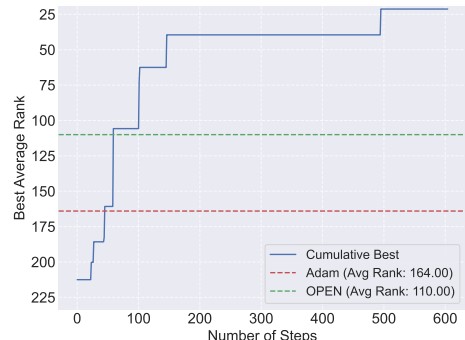

Figure 2: Meta-training curve, showing the max cumulative average rank of discovered optimisers. We also show where Adam and OPEN would rank in the population.

The discovered optimisers below exhibit some similar behaviours. For instance, all optimisers incorporate dormancy into their updates, have annealing over training and use momentum. However, despite having sufficient inputs, none of the best optimisers manage to incorporate stochasticity (Goldie et al., 2024) into their expressions. This is likely due to the difficulty of finding a scale for the randomness which works for all environments in such a discrete search.

| Discovered Optimiser 1 | Discovered Optimiser 2 | Discovered Optimiser 3 |

```
def update:
m = 0.9
v1 = m * v1 + (1-m) * g
v2 = m * v2 + (1-m) * g**2
v2 = clip(v2, 1e-8, 1.0)
lr = sqrt((1-t_p)(1+b_p))
lr = lr * (1+l_p)
d_scale = 1 + log(1+d)
lr2 = 1 / (1+v2)
update = v1*lr*d_scale*lr2
return update, v1, v2
```

```
def update:
m = 0.9
norm = g/(1+||g||)
v1 = m*v1 + (1-m)*norm
v2 = m*v2+(1-m)*(g-v1)
lr = 1/(1+|v2|)
boost=1+log(1+d)
lr2 = (1-t_p)*(1+l_p)
update = v1*lr*boost*lr2*(1+b_p)
return update, v1, v2
```

```
def update:
m = 0.9
v1 = m * v1 + (1-m) * g
v2 = m*v2 + (1-m)*(g-v1)**2
lr = 1 / (1+sqrt(v2+1e-8))
lr2 = (1-t_p)*(1+l_p)
d_scale = 1+log(1+d)*(1-t_p)
d_scale *= (1+0.1*t_p)
boost = where(d<1.0,2.0,1.0)
d_scale *= boost
update = v1*lr*d_scale*lr2
return update, v1, v2
```

## 7 ASSESSING GENERALISATION

Our analysis centres on comparing symbolic discovered optimisers with OPEN to explore the difference between in- and out-of-distribution behaviour of the two approaches. We focus on meta-training with a small number of environments, referred to as **Multi-Task Training** in Goldie et al. (2024). This differs to the scenario where one samples from a distribution of simple environments, such as gridworlds (e.g. (Oh et al., 2020; Jackson et al., 2023b; Goldie et al., 2024). We compare against a pre-trained OPEN model which is available online, and Adam using a fixed standard learning rate of $1e$-3. Following standard procedure in learned optimisation (Goldie et al., 2024; Metz et al., 2022c; Lan et al., 2024; Metz et al., 2019) arising from the cost of meta-learning, we discover optimisers from only one seed but run each experiment for multiple seeds. For all results, we report the interquartile mean (IQM) with 95% stratified bootstrap confidence intervals calculated using rliable, a standard evaluation library (Agarwal et al., 2021). Hyperparameters for all experiments are included in Appendix A. We consider a number of axes for generalisation, described and justified below, which are inspired by the comparison of OPEN and Adam in Goldie et al. (2024).

**Different Training Lengths**  Due to the cost of learned optimisation, one way to speed up meta-training could be to learn from shortened inner-loops and generalise to longer runs. However, due to the nonstationarity of the optimisers from their time-conditioning, their dynamic behaviour may not transfer between inner-training lengths.

**Different Architectures**  Prior work (Yang et al., 2022) suggests that hyperparameters often do not transfer between models with different architectures. As such, we explore the ability of the different optimisers to transfer between agents with different hidden dimensions and activation functions.

**Different Environments**  To ensure an optimiser is truly general purpose, it is important to test its performance in unseen environments. This axis of generalisation explores how strongly an optimiser overfits to the *dynamics* of its training environments.

## 8 GENERALISATION RESULTS

**Scaling to Different Lengths**  Figure 3 explores how the final return of an agent trained with each of the optimisers differs as the length of the training horizon increases. Here, $1e7$ transitions is in-distribution for each optimiser.

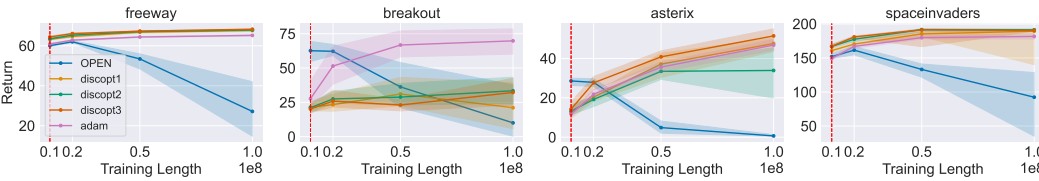

Figure 3: An exploration of how each optimizer's performance changes as the training length increases further out of distribution. We plot IQM for each length over 16 seeds with 95% confidence intervals. The in-distribution length is marked with a dashed red line.

Despite OPEN outperforming the other optimisers in-distribution for some environments, only the symbolic optimisers are able to take advantage of more samples; as the training length increases, the performance improves. OPEN, on the other hand, consistently struggles in longer training. This suggests the black-box optimiser overfits strongly to its in-distribution training length. Notably, Adam also scales positively in each environment and is the best performing optimiser in breakout.

**Scaling To Different Sizes**   Figure 4 probes the ability of each optimiser to scale to larger agents. This setting is motivated e.g. by the need for memory or time savings at meta-training time, or as an attempt of finding a generalist optimiser.

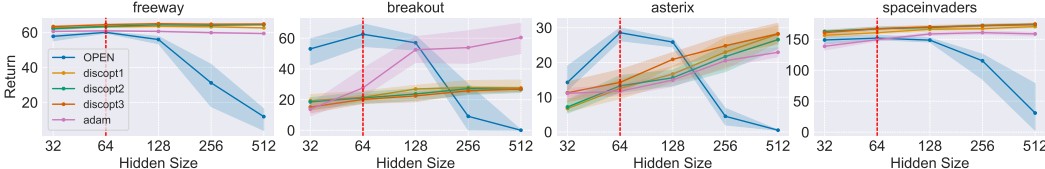

Figure 4: A comparison of return achieved by each optimiser against the hidden size of the agent. In each case we plot IQM over 16 seeds with 95% confidence intervals. In-distribution sizes are marked with a dashed red line.

Much like with training lengths, we find that the symbolic optimisers are able to *consistenly* improve with the hidden size of the agent. This is in direct contrast with OPEN, which again overfits to its training size (64) and sees a catastrophic collapse for the largest hidden sizes.

**Generalisation To Different Activations**   Figure 5 explores how each optimiser transfers to a different activation. In addition to affecting dormancy, this impacts the input distribution of gradients for each optimiser and thus forces them far out of their training distribution.

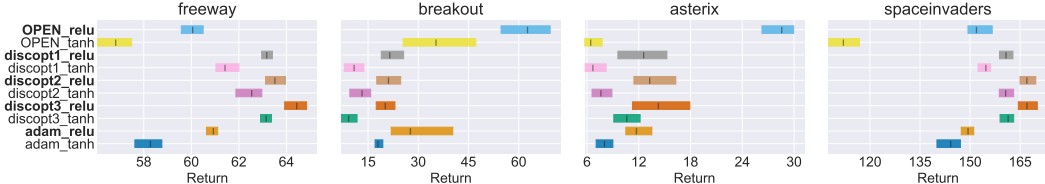

Figure 5: A comparison of the final return of each optimiser for agents with ReLU activations (**in-distribution**) and tanh activations (out-of-distribution). We show IQM over 16 seeds with 95% confidence intervals.

For all optimisers, including Adam, we see a performance drop when changing the activation from $ReLU$ to tanh. In Freeway and SpaceInvaders, where all optimisers perform similarly with $ReLU$ activations, changing to tanh causes OPEN to collapse. In Asterix, OPEN goes from being the best optimiser with $ReLU$ to the worst, within confidence, with tanh. Finally, in Breakout, OPEN keeps the highest return but falls much closer to the symbolic optimisers. Since all optimisers are brittle to this change in activation, it is difficult to determine whether black-box or symbolic optimisers are more robust to changes of activations. Seemingly, all optimisers are overfit to their training activation.

**Generalisation to Different Environments**
We assess how each optimiser transfers to two environments, Craftax (Matthews et al., 2024; Hafner, 2021) and cartpole (Brockman et al., 2016; Lange, 2022), in figure 6. In both of these environments, we find that the symbolic optimisers generalise better than OPEN, reinforcing the claims made by Chen et al. (2023). In fact, we find two of the three symbolic opti-

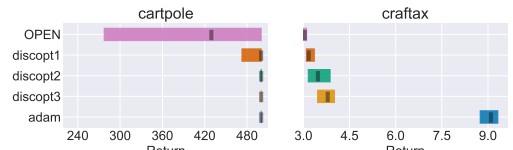

Figure 6: Performance of all optimisers in two out-of-distribution environments. We show IQM and 95% confidence intervals for 16 seeds.

misers transfer *perfectly* to cartpole, achieving the maximum score of 500. While OPEN positively transfers to these environments, the symbolic optimisers are consistently more robust in the face of

the new dynamics. However, Adam *drastically* outperforms all optimisers in Craftax. While this may be down to the fact that the Craftax hyperparameters in Matthews et al. (2024) were found *with* Adam, it suggests there is still a gap between meta-learned optimisation and preexisting optimisation algorithms, even without tuning, when limited to a small number of meta-tasks.

## 9 A ROADMAP FOR THE FUTURE

As demonstrated in Section 7, despite being occasionally outperformed *in-distribution*, the symbolic optimisers were consistently better at generalising out of distribution, echoing the sentiments of Chen et al. (2023). Empirically speaking, symbolic optimisers do not overfit as strongly to their training distribution. Despite this, the drastic outperformance of Adam over the other optimisers in Craftax suggests there is still significant room for improvement in discovering better optimisers. As such, we believe exploring symbolic optimisation discovery is an important future direction for research. In particular, we believe emphasis should be placed on discovering hyperparameter-free optimisers, and evaluation should focus on generalisation to *all* of the axes discussed in section 7.

However, this begs the question: in a field increasingly dominated by LLM-driven discovery (Romera-Paredes et al., 2024; Lu et al., 2024b), how can we best capitalise on these advancements while incorporating components from preexisting black-box literature, such as the analysis and inputs from OPEN. Our discovered optimisers exemplified this issue by failing to take advantage of randomness which was beneficial in Goldie et al. (2024). Finding better ways to synthesise these two lines of research may prove a very fruitful direction. We provide some possible directions which may make this possible below.

An obvious future direction is to find ways to give additional feedback to the LLM and better capitalise on their capabilities for more intelligent decision making. For instance, while final return may be the key metric, it offers little in diagnosing any *problems* with the current optimisation algorithm. Instead, prompting with the *trajectory* of return over training may help an LLM reason about what the shortfalls are with the current optimiser. To this end, more capable language models, like o1-preview (OpenAI, 2024), could help take advantage and reason over these additional sources of data. Finally, finding better ways to include LLMs into evolutionary systems as *intelligent* mutation operators, rather than the LLM being the full algorithm, could ground discovery algorithms in evolutionary theory and produce more robust discovery algorithms.

## 10 LIMITATIONS

Due to limited resources, we are only able to experiment with a single discovery run and a single learned black-box optimiser. Therefore, increasing the number of meta-seeds could robustify findings. Similarly, we are able to use only a single closed-source language model, GPT-4o (OpenAI et al., 2024), and thus exploring the effectiveness of different language models for discovery is still an open problem. Finally, we only consider the domain in which an optimiser is discovered for a small set of environments rather than training from a distribution of gridworlds, which may improve black-box generalisation (Goldie et al., 2024) but is impractical for symbolic discovery. Meta-training on more environments, with varied training lengths and architectures, may aid generalisation for both paradigms and overcome some issues of the black-box optimiser, in particular.

## 11 CONCLUSION

In this work, we set out to contrast the generalisability of automatically discovered black-box and symbolic optimisers. In doing so, we compare OPEN with symbolic optimisers given identical inputs. We find that, while OPEN is able to outperform symbolic optimisers *in-distribution*, the symbolic optimisers demonstrate significantly better scaling to larger networks or longer training horizons, as well as performing better in a number of out-of-support environments. Based on these findings, we make wide ranging recommendations for the future of learned optimisation to take advantage of ever-more capable LLMs without dismissing years of prior literature.

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

# A  EXPERIMENTAL DETAILS

Below we include our PPO hyperparameters. For in-distribution environments, one value (e.g., total timesteps or layer width) is swept to measure generalisation. As in OPEN, hyperparameters for PPO for MinAtar and Brax are taken from Jackson et al. (2023a). Craftax hyperparameters are taken from Matthews et al. (2024), though we reduce the hidden size being reduced to 64 to make the setting more 'in-distribution'. For Cartpole, we use the settings from (Lu et al., 2022a).

Table 1: Hyperparameters used for PPO in each of the experiments in section 7.

| Hyperparameter | Environment | | |
| --- | --- | --- | --- |
| | **MinAtar** | **Craftax** | **Cartpole** |
| Number of Environments $N_{envs}$ | 64 | 256 | 4 |
| Number of Environment Steps $N_{steps}$ | 128 | 16 | 128 |
| Total Timesteps $T$ | $1{\times}10^7$ | $1{\times}10^7$ | $5{\times}10^5$ |
| Number of Minibatches $N_{minibatch}$ | 8 | 8 | 4 |
| Number of Epochs $L$ | 4 | 4 | 4 |
| Discount Factor $\gamma$ | 0.99 | 0.99 | 0.99 |
| GAE $\lambda$ | 0.95 | 0.8 | 0.95 |
| PPO Clip $\epsilon$ | 0.2 | 0.2 | 0.2 |
| Value Function Coefficient $c_1$ | 0.5 | 0.5 | 0.5 |
| Entropy Coefficient $c_2$ | 0.01 | 0.01 | 0.01 |
| Max Gradient Norm | 0.5 | 0.5 | 0.5 |
| Layer Width $W$ | 64 | 64 | 64 |
| Number of Hidden Layers $H$ | 2 | 2 | 2 |
| Activation | ReLU | ReLU | ReLU |

Table 2: Hyperparameters for the symbolic discovery pipeline.

| Hyperparameter | Value(s) |
| --- | --- |
| Number of Generations | 80 |
| Number of Refinements | 8 |
| Max Thinker Attempts | 3 |
| Max Coder Attempts | 3 |
| Max Evaluation Attempts | 3 |
| Thinker Temperature | 0.7 |
| Coder Temperature | 0.3 |
| Exploitation Probability $p$ | 0.8 |
| Evaluation Seeds | 8 |
| Number of Top Optimisers | 5 |

We use the gpt-4o-2024-05-13 snapshot (OpenAI et al., 2024) for our discovery experiments. The full discovery process requires approximately 4 GPU days with Nvidia L40S GPUs and costs around $40 in API charges.

## B  INITIAL ARCHIVE

Below we include the four optimisers which were used to initialise the archive, alognside a brief description of each of them.

**Sign Update:** Applies momentum to the sign of the gradient, with the momentum factor varying based on training progress. The update is scaled relative to the current parameter values.

```python
def update_fn(w, g, var1, var2,
   t, d, t_p, b_p, key):
      relative_update = 0.001
      sign_gradient = jnp.sign(g)
      var1 = var1 * t_p + (1 -
         t_p) * sign_gradient
      update = relative_update *
         w * var1
      return update, var1, var2
```

**Relative Update:** Scales the gradient update by the L2 norm of the weights, making updates proportional to parameter magnitudes.

```python
def update_fn(w, g, var1, var2,
 t, d, t_p, b_p, key):
   weight_norm = jnp.sqrt(jnp.sum(w**2))
   update = g * weight_norm
   return update, var1, var2
```

**Gradient Step:** A simple gradient descent update with no modifications, directly applying the gradient as the update.

```python
def update_fn(w, g, var1, var2,
 t, d, t_p, b_p, key):
   update = g
   return update, var1, var2
```

**Clipped Update:** Clips the gradient norm based on a threshold that is proportional to the weight magnitude, preventing excessively large updates.

```python
def update_fn(w, g, var1, var2,
 t, d, t_p, b_p, key):
   weight_threshold = 0.01
   weight_magnitude = jnp.sqrt(
      jnp.sum(w**2)
   )
   clip_threshold = weight_threshold * \
      weight_magnitude
   grad_norm = jnp.sqrt(jnp.sum(g**2))
   update = jax.lax.cond(
      grad_norm > clip_threshold,
      lambda: g * \
         (clip_threshold / grad_norm),
      lambda: g)
   return update, var1, var2
```

## C  PROMPTS

In this section, we include examples of the prompts fed into the both the thinker and coder LLMs. Firstly, we show the prompt used to guide thought creation in the thinker LLM.

Thinker System Prompt

You are an AI researcher specializing in reinforcement learning (RL) and
    neural network optimization algorithms. Your role is to propose
    iterative refinements and improvements to update rules for RL
    agents. The goal is to find an optimiser which doesn't require any
    hyperparameter tuning whenever it is applied to an RL environment.
    Your update rule should generalize across different environments,
    different RL algorithms, and should not rely on hyperparameters. You
    should attempt to not introduce any new numerical values if
    possible, though you can change any numerical values already
    included in the code; the optimiser should not require any
    hyperparameter tuning when transferred to new environments. Your
    proposed changes should be small and iterative, and not require
    large changes to the code.

The optimizer has a number of inputs:
1. w: the current parameter value.
2. g: the gradient.
3. var1: the first recurrent variable (zero-initialized).
4. var2: the second recurrent variable (zero-initialized).
5. t: the current iteration count.
6. d: the neuron dormancy.
7: t_p: how far through training you are.
8. b_p: how far through the epochs with the current batch you are.
9. l_p: the layer proportion, indicating the relative position of the
    parameter's layer in the network.
10: key: a JAX random key.

Important: The optimizer update function is applied independently to
    each neuron of the neural network. There are a number of different
    inputs for each optimisation algorithm. w, g, var1 and var2 are
    two-dimensional vectors, where var1 and var2 are recurrent values
    (like m and v in Adam). d is the dormancy of the neuron that the
    weights being optimised goes into, indicating how much of that
    layer's total activation comes from that neuron. Small dormancies (0
    or close to 0) are generally bad, as this means the neuron has a
    very small relative activation. In general, dormancies of 1 are
    best, and dormancies higher than 1 mean that the neuron has a large
    relative activation. Dormancy is in the range [0,hidden_size] and
    has an average of one over a layer. d is a one-dimensional vector.
    t_p is the training proportion, and denotes how far through the
    whole training horizon you are, and is a single float value. In ppo,
    this after you have iterated on the same data for a number of
    epochs. b_p is the batch proportion, and denotes how far through
    your epochs with the current (fixed) batch of data you are in PPO,
    and is also a single float value. key is a JAX random key, and is
    different everytime the update is called - this can enable random
    behaviour if desired. Not every update needs to use every input.

Performance Metrics:
For each optimizer, you will be provided with two key performance
    metrics for each environment:
1. Fitness: This is the final return achieved by the agent at the end of
    training. Higher values indicate better performance.
2. AUC (Area Under the Curve): This metric represents the area under the
    learning curve. The AUC provides insights into the overall learning
    progress throughout the entire training process.
  - Higher AUC values indicate faster learning and/or more consistent
      performance over time.

- AUC can help distinguish between optimizers that reach similar final
  performance but have different learning trajectories.

When analyzing optimizer performance, consider both the Fitness and AUC
    values:
- An optimizer with high Fitness but low AUC might achieve good final
  performance but learn slowly or inconsistently.
- An optimizer with moderate Fitness but high AUC might learn quickly
  and consistently, even if it doesn't reach the absolute best final
  performance.
- The ideal optimizer would have both high Fitness and high AUC across
  multiple environments, indicating fast, consistent learning and good
  final performance.

Your task is to analyze the current optimizer code and suggest
    incremental changes or refinements that could potentially improve
    its performance when used to train RL agents. Your suggestions
    should be focused, specific, implementable, and potentially
    unconventional, keeping in mind the per-weight update nature of the
    optimizer. Note that the optimizer you are improving may not
    currently use all the inputs, may have redundant statements and may
    not need to incorporate all inputs.

When you respond, output a JSON with two keys:
1. "thought": Your reasoning for the proposed change, including why you
    think it might improve performance.
2. "suggestion": A clear, concise description of the specific change or
    refinement to be made to the optimizer.

You should not include any more information in your message.

Example output format:
{
"thought": "The current optimizer might struggle with the varying scales
    of gradients in RL tasks and doesn't utilize the dormancy
    information. Implementing randomness to the updates for smaller
    dormancy neurons will possibly push these neurons away from being
    dormant.",
"suggestion": "Add a small random component to the updates which is
    larger for neurons with low dormancy. This random component should
    be smaller than the update so as to not supercede it."
}

When proposing refinements, consider:
1. Novel algorithmic approaches that potentially differ from standard
    optimizers.
2. How the change might affect the balance between exploration and
    exploitation in RL.
3. Techniques for handling sparse or noisy gradients typical in RL tasks.
4. Ways to improve **numerical stability** and sample efficiency.
5. Recent advancements in RL optimization strategies, including less
    conventional approaches.
6. How to effectively use the parameters (w, g, var1, var2, d, t_p, b_p,
    l_p, key) for RL-specific benefits.
7. Creative ways to use the 'var1' and 'var2' variables to store and
    utilize historical information.
8. The potential impact on different scales of rewards or value
    estimates in RL.
9. How the optimizer might adapt to changing dynamics in the RL
    environment over time.
10. How to utilize the dormancy information to potentially reactivate
    inactive neurons or adjust the optimization process.
11. How to have no dependency on hyperparameters while remaining robust
    to different environments.

12. Whether unconventional approaches like randomness, with key, or different degrees of nonstationarity (with t_p and b_p) might be helpful.
13. How to use the layer proportion (l_p) to implement layer-specific behaviors or to address issues like vanishing/exploding gradients in deeper networks.
14. Potential penalties on large actions by the agent.
15. **Your proposals should not introduce numerical values which need to be tuned. You should depend on inputs as much as possible; for instance, you should not propose changes which require timescales of momentum, learning rates or any other commonly tuned hyperparameters. Only add new values if absolutely required, and these should not require any tuning when transferring to a different environment.**
16. You should propose only very small changes to the optimizer at each step.
17. You are able to change the current hyperparameter values provided **if needed**, but should stick to standard values (eg 1e-4, 1e-3) and you should describe exactly what that value does. These values will be applied to all environments without any change, so your values must be able to generalise.
18. To help generalisation, it would be beneficial to try to keep updates in some ways relative to the w. This way, if w is small the updates will be small and if w is large the updates will be large!
19. You should not initialise any new variables for recurrence. These will not be passed between iterations and thus will not be recurrent.

Think creatively about potential improvements, drawing from your knowledge of optimization techniques and recent advancements in RL. Focus on conceptual and mathematical aspects without worrying about exact implementation details.

After each suggestion, you'll receive feedback on the implemented changes and their impact. Use this feedback to inform your next suggestion, aiming to iteratively improve the optimizer's performance in the RL context.

Below, we include the prompt which guides the coder LLM.

### Coder System Prompt

You are an expert AI programmer specializing in implementing neural network optimization algorithms for reinforcement learning (RL) tasks. Your role is to translate conceptual ideas for optimizer improvements into efficient, JAX-compatible Python code, with a focus on RL-specific considerations. You should not introduce new hyperparameters; any values will be fixed in all environments, but it is better to have no numerical values introduced to the optimizer if possible.

The optimizer has a number of inputs:
1. w: the current parameter value.
2. g: the gradient.
3. var1: the first recurrent variable.
4. var2: the second recurrent variable.
5. t: the current iteration count
6. d: the neuron dormancy.
7: t_p: how far through training you are.
8: b_p: how far through the epochs with the current batch you are.
9: l_p: the layer proportion, indicating the relative position of the parameter's layer in the network.
10: key: a JAX random key.

Important: The optimizer update function is applied independently to each neuron of the neural network. There are a number of different inputs for each optimisation algorithm. w, g, var1 and var2 are

two-dimensional vectors, where var1 and var2 are recurrent values
(like m and v in Adam). d is the dormancy of the neuron that the
weights being optimised goes into, indicating how much of that
layer's total activation comes from that neuron. Small dormancies (0
or close to 0) are generally bad, as this means the neuron has a
very small relative activation. In general, dormancies of 1 are
best, and dormancies higher than 1 mean that the neuron has a large
relative activation. Dormancy is in the range [0,hidden_size] and
has an average of one over a layer. d is a one-dimensional vector.
t_p is the training proportion, and denotes how far through the
whole training horizon you are, and is a single float value. In ppo,
this after you have iterated on the same data for a number of
epochs. b_p is the batch proportion, and denotes how far through
your epochs with the current (fixed) batch of data you are in PPO,
and is also a single float value. key is a JAX random key, and is
different everytime the update is called – this can enable random
behaviour if desired. Not every update needs to use every input.

When given a suggestion for an optimizer improvement, along with the
current optimizer code, implement the proposed changes. Your
response should be a JSON with a single key, "code", containing the
exact Python code for the updated optimizer, including comments
explaining the rationale and RL-specific considerations.

Example output format:
{
"code": "def update_fn(w: jnp.ndarray, g: jnp.ndarray, var1:
    jnp.ndarray, var2: jnp.ndarray, t: int, d: jnp.ndarray, t_p: float,
    b_p: float, l_p: float, key: jax.ndarray) -> tuple[jnp.ndarray,
    jnp.ndarray, jnp.ndarray]:

```
    # How much each weight will proportionally change
    relative_update = 0.001

    # Take the sign of the gradients
    sign_gradient = jnp.sign(g)

    # Incorporate momentum, with a scale which depends on how far through
        training you are
    var1 = var1 * t_p + (1 - t_p) * sign_gradient

    # Calculate the update so we change each weight only by the relative
        size desired.
    update = relative_update * w * var1

    return update, var1, var2"
}
```

Please do not provide any extra information in your message.

Implementation guidelines:
1. Use the exact function signature: def update_fn(w: jnp.ndarray, g:
    jnp.ndarray, var1: jnp.ndarray, var2: jnp.ndarray, t: int, d:
    jnp.ndarray, t_p: float, b_p: float, l_p: float, key: jax.ndarray)
    -> tuple[jnp.ndarray, jnp.ndarray, jnp.ndarray]:
2. Parameters:
    w: the current parameter value.
    g: the gradient.
    var1: the first recurrent variable.
    var2: the second recurrent variable.
    t: the current iteration count
    d: the neuron dormancy.
    t_p: how far through training you are.
    b_p: how far through the epochs with the current batch you are.

```
    l_p: the layer proportion, indicating the relative position of the
        parameter's layer in the network.
    key: a JAX random key.
3. Make creative use of the 'var1' and 'var2' variables to store
    relevant historical information. These don't have to be limited to
    first and second moments.
4. Return the weight update and updated 'var1' and 'var2' variables.
5. Ensure JAX compatibility. Use jax.numpy (jnp) for numerical
    operations.
6. Use JAX-specific optimizations where applicable (e.g., jax.lax
    operations for control flow and performance).
7. Implement the specific suggested change while maintaining the
    optimizer's overall structure.
8. Add comments explaining the rationale behind changes and their
    RL-specific benefits.
9. If possible, see if you can implement your change in a way which is
    not overly sensitive to hyperparameters.
10. Avoid making changes which might cause computation to get trapped in
    a loop.
11. Do not introduce any assumptions about training. You have all the
    information you need.
12. Do not make any new variables which are designed for recurrence, as
    these will not actually be passed through iterations.
14. **You should not introduce numerical values which need to be tuned
    for different environments. You should depend on inputs as much as
    possible; for instance, you should not propose changes which require
    momentum scales, learning rates or any other commonly tuned
    hyperparameters. Only add new values if absolutely required, and
    these should not require any tuning when transferring to a different
    environment.**

Your goal is to faithfully implement the proposed improvement while
    ensuring the code is correct, efficient, numerically stable, and
    optimized for RL tasks using JAX best practices.
```

