# OpenReview forum: "Do Symbolic or Black-Box Representations Generalise Better In Learned Optimisation?"
_ICLR.cc/2025/Conference — Submitted to ICLR 2025_

### Official Review · Reviewer_BFLw · 2024-10-28

**Soundness:** 2
**Presentation:** 2
**Contribution:** 1
**Rating:** 3
**Confidence:** 4

**Summary:**

The paper presents a method leveraging Large Language Models (LLMs) to improve symbolic optimizers for reinforcement learning (RL) sequentially. It addresses the generalization abilities of symbolic versus black-box optimizers in learned optimization, where the goal is to automate the development of optimization algorithms that perform effectively across varied environments. Specifically, the paper builds a pipeline that discovers hyperparameter-free symbolic optimizers, facilitating a comparison with black-box methods, particularly the open-source black-box optimizer OPEN. The authors run evaluations on 4 datasets and the generalization to out-of-distribution datasets, and other hyparapemters (such as activations, sizes, and training lengths).

**Strengths:**

* The paper addresses a crucial issue in RL optimization by exploring which type of optimizer—symbolic or black-box—offers better generalization. This topic is particularly relevant as RL optimization algorithms are costly to meta-learn and difficult to generalize across environments.
*  The background and related work sections provide a well-rounded introduction, grounding readers in the context of learned optimization and its challenges in RL.
*  The paper offers a thoughtful discussion of results and future research directions to improve symbolic optimizers and integrate findings from black-box optimization literature.

**Weaknesses:**

* The approach relies on a specific LLM, GPT-4o, making the results dependent on the capabilities and limitations of that model. This dependency may reduce the robustness and reproducibility of the approach, as outcomes may differ with other LLMs. Moreover, the dependency on LLMs makes this idea difficult to scale.

* The study uses only one black-box optimizer (OPEN) and limited baselines, evaluating across only four RL environments. The restricted dataset and baselines' choice may limit the findings' generalizability and impact.

* A core issue in meta-learning is achieving strong performance across diverse tasks. While symbolic optimizers demonstrate some benefits, they struggle in certain out-of-distribution tasks, underscoring the primary challenge the paper seeks to address. Additional insights on generalization could further strengthen the work.

* The paper contains numerous typos and moments of complex or unclear writing, making it occasionally challenging to follow. Moreover, more than half of the paper is spent discussing background, motivation, and future directions, while experimental sections, crucial for substantiating claims, could be expanded and discussed in greater depth.

**Questions:**

* The authors assume that the results on open-source models will be equally good. Is there any solid empirical evidence for this?
* Why the optimiser does not transfer to new tasks?
* Is there a possibility of overfitting? How can the current model avoid overfitting?

---

> ### Author Response · Authors · 2024-11-25
>
> Dear BFLw,
>
> Thank you for your feedback and review. We are glad that you appreciate the research question posed in this paper, and our well-structured motivation and background section. Below, we answer the questions raised in your review.
>
> >The approach relies on a specific LLM, GPT-4o, making the results dependent on the capabilities and limitations of that model. This dependency may reduce the robustness and reproducibility of the approach, as outcomes may differ with other LLMs.
>
> While we used GPT-4o for our experiments, we specifically designed our discovery pipeline to be robust across different LLMs. Several aspects support this:
>
> 1. A two-LLM setup (thinker/coder) with deliberately constrained prompts that focus on targeted incremental changes rather than relying on sophisticated reasoning. This modular approach reduces dependency on high-end LLM capabilities.
> 2. The pipeline builds on established evolutionary principles, using LLMs primarily as intelligent mutation operators rather than having them drive the full discovery process. This grounding in evolutionary computation provides stability even with simpler LLMs.
>
> While validating across multiple LLMs would strengthen our findings (noted in Limitations), our core contribution of comparing symbolic vs black-box optimization remains valuable and reproducible.
>
> >The restricted dataset and baselines' choice may limit the findings' generalizability and impact.
>
> While we are only able to learn from one meta-training distribution due to computational constraints, we believe that these findings should generalise beyond the scope of this paper. As noted in Section 5, we choose to consider the "Multi-Task Training setting... where there is a small number of meta-tasks" (line 238) as this represents a scenario often faced, where a distribution of environment is not well-defined (such as if trying to learn for a suite of games). Whilst it would be beneficial to explore learning in a distribution of environments, which was beneficial for generalisation in [1], this is poorly defined for symbolic discovery with a language model and thus has been left open to future work.
>
> >they struggle in certain out-of-distribution tasks, underscoring the primary challenge the paper seeks to address.
>
> It is important to note that this paper is not proposing a method to address these issues, but instead looks to analyse the difference between black-box and symbolic algorithm discovery pipielines. The fact that there is still a struggle to transfer to different out-of-distribution tasks is not a limitation of this work, but a demonstration that even a supposedly 'more generalisable' method still has limitations. We intend for this work to provide a platform to those looking to pursue algorithm discovery in the future by offering an idea of which avenue might be more fruitful.
>
> >The paper contains numerous typos and moments of complex or unclear writing, making it occasionally challenging to follow.
>
> We apologize for any moments of unclear writing or typos that impacted readability. We have corrected all mentioned typos and grammatical errors in red-line markup on the revised paper. Due to the large amount of background relevant to this field, we feel it is appropriate to highlight the necessary related work. Similarly, since the goal of the paper is to analyse promising future research avenues in this field, the inclusion of future work should be considered as central to the paper rather than incidental.

---

> > ### Author Response · Authors · 2024-11-25
> >
> > ---
> >
> > >The authors assume that the results on open-source models will be equally good. Is there any solid empirical evidence for this?
> >
> > We acknowledge that Section 5's statement about system robustness with weaker models would be strengthened by empirical validation. We attempted to validate this claim with several open-source models but encountered technical difficulties and computational constraints that prevented us from completing a formal comparison within the submission timeline, though preliminary tests suggested this claim would be substantiated. We will run necessary experiments in time for the camera-ready version of this paper.
> >
> > However, several aspects of our pipeline's design suggest transfer:
> >
> > 1. Our thinker/coder separation deliberately constrains the required capabilities to basic code manipulation and mathematical reasoning, avoiding dependence on sophisticated abilities unique to high-end LLMs.
> > 2. The mutations proposed are small and incremental rather than requiring complex reasoning chains or creative leaps. Most modern open-source LLMs demonstrate competence at such targeted code modifications.
> >
> >
> >
> > >Why the optimiser does not transfer to new tasks?
> >
> > It is likely that either the optimisers are not sufficiently expressive and adaptive to respond to the different input distributions. However, it is important to note that this work focuses on *analysis* rather than proposing a new optimisation algorithm, and as such the lack of transfer has formed part of our analysis - it is an insight rather than a limitation of this work. At the same time, there are (thus far) no optimisation algorithms which can generalise 0-shot to all conceivable tasks.
> >
> >
> >
> > >Is there a possibility of overfitting? How can the current model avoid overfitting?
> >
> > This is an interesting insight. Part of the basis for our development of this research question is whether simple, symbolic algorithms do not overfit the meta-training tasks as much as black-box ones. This may be a contributor to why they seem to generalise better. That said, it is likely that there is still a level of overfitting that limits algorithmic transfer, and therefore we hope that this work provides a platform to future research in this space.
> >
> >
> >
> > [1] Can Learned Optimization Make Reinforcement Learning Less Difficult? Alexander D. Goldie, Chris Lu, Matthew Thomas Jackson, Shimon Whiteson, Jakob Nicolaus Foerster. 2024.

---

> > > ### Comment · Reviewer_BFLw · 2024-11-26
> > > **Reply to authors**
> > >
> > > I thank the authors for their response. However, we are not convinced and still thinks. I acknowledge I read the rebuttal but decide to keep my score.
> > >
> > > I summarize my remaining concerns as:
> > >
> > > * More baselines, datasets, and LLMs should be tried to make the evidence stronger.
> > > * The generalization to other tasks should be important, otherwise there will be limited benefits.

---

### Official Review · Reviewer_KBh5 · 2024-10-30

**Soundness:** 2
**Presentation:** 2
**Contribution:** 2
**Rating:** 3
**Confidence:** 4

**Summary:**

This paper proposes a symbolic discovery pipeline which prompts a LLM-based thinker and a LLM-based coder to iteratively produce new optimizers based on an optimizer archive for RL training. Experimental results show that the discovered optimizers outperform Adam and OPEN on the training RL tasks.

**Strengths:**

1. The presentation in the paper is very clear.

2. The proposed LLM-based algorithm generation pipeline needs less hyper-parameter and expertise.

**Weaknesses:**

1. The pipeline requires a large number of LLM conversations and RL inner training loops, which is time and resource consuming.

2. The compared baselines are limited, in the experiment authors only compare two baselines, OPEN and Adam, advanced baselines such as SGD, Lion and AdamW are not included, the performance differences between the discovered optimizers and the optimizers in the archive are also omitted.

3. The generalization performance of the discovered optimizers does not significantly outperform Adam, especially in the generalization to different environments.

**Questions:**

1. How to select optimizers for the archive, and why are these 4 optimizers chosen? Will archive with more or less optimizers affect the performance? Will optimizers with different implementations affect the searching?

2. The codes generated by LLMs may not always be correct, is there any code review and recover (if errors occur) mechanisms?

3. What is the adventage of using 2 LLMs as thinker and coder, instead of using a single LLM to directly generate the optimizer? How thinker LLM help coder LLM to obtain better optimizer?

4. Can the discovered optimizers generalize to other RL methods (i.e., DQN or REINFORCE) or non-RL learning paradigms (i.e., supervised learning), since the experiments only include PPO paradigm?

---

> ### Author Response · Authors · 2024-11-25
>
> Dear KBh5,
>
> Thank you for your review. We appreciate your positive response to our LLM-driven discovery pipeline. Below, we respond to some specific aspects of your review.
>
> > requires a large number of LLM conversations and RL inner training loops, which is time and resource consuming.
>
> While this is true, this is a fact for any type of meta-learned algorithms in reinforcement learning. This is actually the underpinning of our research question: since the cost of algorithm learning or discovery is very high, it should emphasise generalisation to have benefits at meta-test time. Due to our emphasis on hyperparameter-free algorithms, our hope is that the findings from this work will help to guide the field of algorithm discovery and, thus, any upfront meta-training cost will be drastically outweighed by savings down the line in RL training as the field matures.
>
> > baselines are limited, in the experiment authors only compare two baselines, OPEN and Adam
>
> Due to the focus of our paper on comparing symbolic and black-box *discovered* optimisers, we wanted to emphasise the difference in performance by making the comparison between OPEN and the discovered optimisers the focus of our evaluation. This is also the case in other discovered algorithms papers such as [1], where the comparison is *only* between LPG trained with and without unsupervised environment design, or [2], where many comparisons are between only temporally-aware and regular discovered algorithms.
>
> >generalization performance of the discovered optimizers does not significantly outperform Adam
>
> Whilst this is true, as above our emphasis was on the difference between black-box and symbolic algorithms rather than between discovered and not. As such, while we include Adam as a point of reference, the focus of our work is on the legitimacy of the claim that 'symbolic algorithms will generalise better than black-box ones' as set out by [3].

---

> > ### Author Response · Authors · 2024-11-25
> >
> > ---
> >
> > >How to select optimizers for the archive, and why are these 4 optimizers chosen? Will archive with more or less optimizers affect the performance? Will optimizers with different implementations affect the searching?
> >
> > We describe our selection process in Section 5.2 Initialisation (line 281). In essence, we chose a small number of initial optimisers which covered some of the behaviour we thought might help discovery down the line, such as relative parameter updates and initialisation to gradient descent. It is likely that increasing the archive will lead to differently performing optimisers, but the hope is that the discovery process provides sufficient coverage and is not too inherently limited by the initial selection. We have been unable to run this additional experiment over the course of rebuttals, but will do so in time for the camera ready version.
> >
> >
> > >The codes generated by LLMs may not always be correct, is there any code review and recover (if errors occur) mechanisms?
> >
> > The system implements a comprehensive error handling and recovery mechanism. Errors are classified into six types (syntax, runtime, naming, numerical, shape, and unknown) and are tracked through ErrorInstance objects that capture detailed information about each error. When errors occur, the system provides specific feedback to the LLMs based on the error type, allowing them to learn from mistakes and improve in subsequent iterations.
> >
> > The system also employs multiple fallback mechanisms for parsing LLM outputs, starting with standard JSON parsing and falling back to regex-based parsing when needed. This multi-tiered approach ensures robust handling of imperfect LLM outputs while maintaining system stability.
> >
> > >What is the advantage of using 2 LLMs as thinker and coder, instead of using a single LLM to directly generate the optimizer? How thinker LLM help coder LLM to obtain better optimizer?
> >
> > We find that for many models, prompting an LLM to propose a change and implement it can lead to unpredictable results, such as code updates that do not match their proposal. We find that separating the process into thinking and coding leads to more predictable behaviour, benefiting not only performance (as the reasoning is linked to the implemented changes) but also interpretability over the discovery process. We have provided additional in-text references to justify this design decision further.
> >
> >
> > >Can the discovered optimizers generalize to other RL methods (i.e., DQN or REINFORCE) or non-RL learning paradigms (i.e., supervised learning), since the experiments only include PPO paradigm?
> >
> > Unfortunately, due to computational restrictions we were unable to experiment with generalisation to new environments. While this would prove valuable as an additional experiment, it is not uncommon to discover algorithms for a specific paradigm (e.g. [1,2]).
> >
> > [1] Discovered Policy Optimisation. Chris Lu, Jakub Grudzien Kuba, Alistair Letcher, Luke Metz, Christian Schroeder de Witt, Jakob Foerster. 2022.
> >
> >
> > [2] Discovering Temporally-Aware Reinforcement Learning Algorithms. Matthew Thomas Jackson, Chris Lu, Louis Kirsch, Robert Tjarko Lange, Shimon Whiteson, Jakob Nicolaus Foerster. 2024
> >
> > [3] Can Learned Optimization Make Reinforcement Learning Less Difficult? Alexander D. Goldie, Chris Lu, Matthew Thomas Jackson, Shimon Whiteson, Jakob Nicolaus Foerster. 2024.

---

### Official Review · Reviewer_JLLy · 2024-11-04

**Soundness:** 2
**Presentation:** 2
**Contribution:** 2
**Rating:** 3
**Confidence:** 4

**Summary:**

To meta-learn RL-specific gradient optimizers, existing work explores two ways, learning a black-box model that outputs parameter updates, or using a symbolic algorithm to discover update equations/codes. This paper first argues that fair comparison among the above two ways is challenging because black-box methods often generalize in a zero-shot manner, while symbolic methods often rely on tunable hyperparameters for each task. Using OPEN [Goldie et al., 2024] as a key baseline, the authors propose a symbolic optimizer discovery pipeline and design a comparison to evaluate which of the above two ways better generalizes in training RL agents. Results indicate that while LLM-based symbolic learning underperforms on in-distribution tasks, it can have better generalization on out-of-distribution ones.

**Strengths:**

* The background and motivation are clearly introduced.
* The LLM-based framework is well-developed.
* The experiments in Section 8, conducted under various conditions/setups, are helpful and valuable.

**Weaknesses:**

* While the paper asks a broad research question regarding the generalization comparison between symbolic and black-box optimizers for learned RL optimizers, its actual study scope is limited, as it primarily compares the proposed LLM-based discovery method with OPEN, a single and particular black-box optimizer. This raises concerns about the generalizability of the conclusions. Additionally, many other methods reviewed and cited in the paper are not included as baselines, which makes the comparison weak.

* Moreover, substantial domain knowledge and successful practices from OPEN are embedded into the prompts (in Appendix C). This reliance makes the discussion highly specific to and dependent on OPEN, also making the method hand-crafted. This seems to diverge from the stated goal of reducing human input, as significant input is now provided through the LLM prompts.

* Running only a single discovery run is indeed a limitation as acknowledged by the authors in Section 10. Also, it is unclear why focusing on a small number of environments during training would be beneficial or not.

* The clarity and organization of this paper could be improved. Providing examples and a clear explanation of what hyperparameters typically represent in the discovered optimizer would be beneficial. Many technical details are either insufficiently explained (e.g., what is meant by “small" relative changes to weights in line 287, what exactly constitutes the multi-task learning setting) or lack sufficient justification (e.g., in line 317, why p=0.8?). Additionally, some details are poorly organized (e.g., technical details of this work in lines 162–165 appear in the related work section rather than in the main methods section).  The “usesw” in line 63 is a typo error.

* More related work can be leveraged and discussed, e.g., [1-2]. Also, the idea of leveraging two LLMs to act as a "thinker" and a "coder" has also been explored in recent LLM-based optimizer discovery studies [3-4].

[1] SYMBOL: Generating Flexible Black-Box Optimizers through Symbolic Equation Learning (ICLR 2024)

[2] Rethinking Branching on Exact Combinatorial Optimization Solver: The First Deep Symbolic Discovery Framework (ICLR 2024)

[3] Evolution of Heuristics: Towards Efficient Automatic Algorithm Design Using Large Language Model (ICML 2024)

[4] ReEvo: Large Language Models as Hyper-heuristics with Reflective Evolution (NeurIPS 2024)

**Questions:**

* Is the “m=0.9” term in the discovered optimizers treated as a fixed hyperparameter?

* Can the discovered optimizers generalize effectively to complex RL tasks, such as continuous-action or high-dimensional control tasks?

* How sensitive are the symbolic optimizer outcomes to changes in prompt design?

* Is the discovery process sensitive to the used meta-training task?

---

> ### Author Response · Authors · 2024-11-25
>
> Dear JLLy,
>
> Thank you for comments and recognition of our clearly motivated problem and well-developed framework. Below, we step through your individual points and look forward to discussing this further throughout the rebuttal process.
>
> >  its actual study scope is limited, as it primarily compares the proposed LLM-based discovery method with OPEN, a single and particular black-box optimizer
>
> > many other methods reviewed and cited in the paper are not included as baselines, which makes the comparison weak.
>
>
> Our goal here was to compare *discovered* symbolic optimisation algorithms with black-box ones, with the emphasis on generalisation; this is particularly relevant in reinforcement learning, where sample complexity is an issue. As such, we made our comparison with the state-of-the-art learned optimisation for reinforcement learning, which is OPEN. We also chose to provide additional results for Adam as context, though this is not nor should it be the principle takeaway from this paper. This method of evaluation, where only a direct comparison is considered, is not uncommon (for example, [1,2]). We provided additional method references to contextualise the field, but these should not be considered part of our empirical comparison.
>
>
> >This reliance makes the discussion highly specific to and dependent on OPEN, also making the method hand-crafted.
>
> Much like OPEN, the level of 'hand-craftedness' is one step removed from the algorithm, meaning the *algorithm* is discovered while the *method* is handcrafted; this is the case in practically all learned algorithms as we are not yet at the capability of automating research itself. Even works like the AI Scientist [3] involve some level of hand-craftedness in how they design the system or allow the LLM to design experiments, for instance.
>
>
> > it is unclear why focusing on a small number of environments during training would be beneficial or not.
>
> OPEN is trained in one of three settings: either in a single environment, on multiple environments or for a distribution of environments. While [4] demonstrated the benefits of training on a distribution for black-box optimisation, there are many domains in which a domain can not be defined - such as when optimising in a game - and thus focus on the setting where we can only train on a small number of 'similar' environments. Though it is not intuitive to train a symbolic optimiser on a distribution of environments, we leave this direction open to future work and have highlighted this as a limitation in our paper.
>
>
>
> >Many technical details are either insufficiently explained ... or lack sufficient justification
>
> Thank you for these suggestions. We have made a number of relevant edits in our new PDF.
>
> > More related work can be leveraged and discussed
>
> Thank you for raising this. We have included the proposed references in relevant locations.

---

> > ### Author Response · Authors · 2024-11-25
> >
> > ----
> >
> > > Is the “m=0.9” term in the discovered optimizers treated as a fixed hyperparameter?
> >
> > Yes, to ensure we maintain the 'hyperparameter-free' component of the discovered optimisers we treat any values as fixed.
> >
> >
> > > Can the discovered optimizers generalize effectively to complex RL tasks, such as continuous-action or high-dimensional control tasks?
> >
> > While our current evaluation focuses on discrete-action environments, our results explore generalization across multiple axes - architecture sizes, training lengths, and environment dynamics (from MinAtar to Cartpole and Craftax). We have run a number of tests on Brax but found stability a significant issue with all optimisers. We have noted in section 10 that additional experimentation with more complex environments would be compelling future work.
> >
> >
> >
> > > How sensitive are the symbolic optimizer outcomes to changes in prompt design?
> >
> > We iterated on our prompt a number of times to attempt to elicit more interesting behaviour. However, we found that performance was not overly sensitive to the prompt design - we hypothesise that this is due to the evolution-inspired system used in our work.
> >
> >
> > > Is the discovery process sensitive to the used meta-training task?
> >
> > Due to cost and computational restraints, we are only able to learn from one meta-training distribution. However, it is likely that the symbolic discovery process is better suited to different meta-training distributions than black-box ones. For instance, generalisation in OPEN particularly came from training on randomly generated distributions of mazes [4] - this process would be poorly defined for symbolic algorithms and, thus, we focused on the 'multi-task training' setup instead.
> >
> > [1] Discovering General Reinforcement Learning Algorithms with Adversarial Environment Design. Matthew Thomas Jackson, Minqi Jiang, Jack Parker-Holder, Risto Vuorio, Chris Lu, Gregory Farquhar, Shimon Whiteson, Jakob Nicolaus Foerster. 2023
> >
> > [2] Discovering Temporally-Aware Reinforcement Learning Algorithms. Matthew Thomas Jackson, Chris Lu, Louis Kirsch, Robert Tjarko Lange, Shimon Whiteson, Jakob Nicolaus Foerster. 2024
> >
> > [3] The AI Scientist: Towards Fully Automated Open-Ended Scientific Discovery. Chris Lu, Cong Lu, Robert Tjarko Lange, Jakob Foerster, Jeff Clune, David Ha. 2024
> >
> > [4] Can Learned Optimization Make Reinforcement Learning Less Difficult?. Alexander D. Goldie, Chris Lu, Matthew T. Jackson, Shimon Whiteson, Jakob N. Foerster. 2024

---

> > > ### Comment · Reviewer_JLLy · 2024-11-29
> > >
> > > I thank the authors for their responses. After reviewing the rebuttal and the comments from other reviewers, I remain concerned about the limited evaluation and discussion provided in this paper. Therefore, I choose to maintain my score of 3.

---

### Official Review · Reviewer_oDvu · 2024-11-04

**Soundness:** 2
**Presentation:** 1
**Contribution:** 3
**Rating:** 3
**Confidence:** 3

**Summary:**

Optimizers are key algorithmic components in machine learning. In addition to man-made optimizers such as Adam, they can be learned automatically with little human guidance, which is the topic of the so called learned optimization. There are two types of learned optimizers: black-box algorithms and symbolic algorithms, in which a black-box optimizer is presented by a neural network while a symbolic optimizer is presented by math or code. For being truly useful, a learned optimizer should be generalized well to unseen settings beyond the ones it was learned. Therefore, it makes sense to study the generalization of the meta-learned optimizers. This manuscript contrasts the generalizability of the auto-learned black-box optimizers and the symbolic optimizers based on some reinforcement learning settings.

The authors use an existing black-box optimizer OPEN as its black-box baseline and they devise a symbolic optimizer discovery pipeline. This pipeline takes an evolutionary paradigm which includes the initialization, selection, mutation, and evaluation phases. What interesting is that it employs LLMs in the mutation phase and split it into the thinker step and the coder step for clearer interpretations.

The authors investigated several aspects as the generalizability, including the training lengths, the model sizes, the activation functions, and the environments. Experimental results show that although the black-box optimizer outperforms the symbolic optimizers in-distribution, the symbolic optimizers surpass the black-box optimizer generally, demonstrating better generalizability.

**Strengths:**

Although the literature claimed that black-box algorithms may be easier to work with while symbolic algorithms may generalize better, there is no strong justification. This manuscript conducts an empirical study to compare these two ideas to justify the assertion.

The designed symbolic optimizer discovery pipeline is inspirational and especially the idea of adopting LLMs is interesting and hot.

**Weaknesses:**

I am concerning about the significance of the learned optimization. As showed in the experiment, Adam outperforms other methods significantly in new environments. Although in the in-distribution settings, Adam performs better or closely to the symbolic optimizers. So what is the value of the learned optimizers since they need much more additional cost to do the meta learning?

Another main weakness of this manuscript is its presentation. It is not so well written and a little difficult to follow. First of all, the research problem is not clearly stated. What is an optimizer and how is it used in machine learning algorithms, at least in RL this manuscript studies? What is the definition of the core concept generalizability in this manuscript? What is the metric and how to evaluate the generalizability? I get these information from the late parts of the manuscript and they should be promoted to the early parts to be reader friendly. Second, a few key terms occur suddenly without any introduction. For instance, the term OPEN firstly appears in line 112 without mentioning before. A reader unfamiliar with this topic would be confused about what it is and where it is from. In the figures, what do the terms discopt1, discopt2, and discopt3 stand for? Third, there are some typos or grammatical errors. I name a few. Line 045: outer-loop -> outer loop; Line 063: usesw -> uses; Line 300: set of -> a set of; Line 358: the full stop is missing; Line 264: they don’t faily catastrophically in the training environments: typo or grammatical error. Anyway, at least regarding the writing and presentation, this manuscript is in a draft state and not ready for publication.

**Questions:**

See the part of weaknesses.

One additional question is that what is the use of the context optimizers in the evolution phase?

---

> ### Author Response · Authors · 2024-11-25
>
> Dear oDvu,
>
> Thank you very much for your feedback. We appreciate that you found our method's interpretability an advantage, as well as the foundations of our research question. We provide a response to your points below, and look forward to an active discussion over the coming days.
>
> > Adam outperforms other methods significantly in new environments
>
> While it is true that Adam is a strong baseline, the point of this work is in comparing the difference between black-box and symbolic algorithms rather than between discovered and not. As such, while we include Adam as a point of reference, the focus of our work is on the legitimacy of the claim that symbolic algorithms will generalise better than black-box ones, as set out by [1], and therefore the focus should be on the relative strenths and weaknesses of the symbolic optimisers with the black-box ones (such as a robustness to out-of-distribution, or performance in-distribution).
>
>
> > what is the value of the learned optimizers since they need much more additional cost to do the meta learning?
>
> The large upfront cost of meta-training is an unfortunate fact for many types of meta-learned algorithms in reinforcement learning. This is part of the importance of our research question; since the cost of algorithm learning or discovery is very high, generalisation needs to be emphasised to have benefits at meta-test time. Due to the focus on hyperparameter-free algorithms, our hope is that the findings from this work will help inform the field of algorithm discovery and, thus, any upfront meta-training cost will be drastically outweighed by savings down the line in RL training as the field matures.
>
>
>
> > It is not so well written and a little difficult to follow.
>
> We apologise for a lack of clarity in certain areas of our manuscript. We have amended grammatical errors, coloured in red, in an updated version which we have uploaded to OpenReview.
>
>
> >what is the use of the context optimizers in the evolution phase?
>
> Thank you for this interesting question. As we describe in our paper (Section 5.3.3 - Previous Performance), we attempt to align our system closely to more conventional evolutionary algorithms despite the usage of a language model for mutation. As such, we include additional context optimisers in the prompt to enable a crossover-like operation, where the language model can incorporate changes which are effective in different environments in the ultimately discovered optimisers. This enables similar behaviour to that described in [2]. Due to the length of conversations, we are unable to include full examples of this effect in our paper.
>
> [1] Symbolic Discovery of Optimization Algorithms. Xiangning Chen, Chen Liang, Da Huang, Esteban Real, Kaiyuan Wang, Yao Liu, Hieu Pham, Xuanyi Dong, Thang Luong, Cho-Jui Hsieh, Yifeng Lu, Quoc V. Le. 2023
>
> [2] Language Model Crossover: Variation through Few-Shot Prompting. Elliot Meyerson, Mark J. Nelson, Herbie Bradley, Adam Gaier, Arash Moradi, Amy K. Hoover, Joel Lehman. 2024

---

> ### Comment · Reviewer_oDvu · 2024-11-27
> **Reply to the authors**
>
> Thanks the authors for their response! I have read the rebuttal and would like to keep my rating. My main concerns are as follows.
>
> (1) Although the authors emphasized that the key motivation of this submission is to compare black-box optimizers and the symbolic optimizers, not to beat the Adam method, the field of learned optimizer has not demonstrated its value and advantage in applications in other problems if it cannot beat the well established Adam method, at least in some settings. I think the core and important problem of this field is to find out a result that can beat Adam at least, not to compare these two types of weak baselines in this submission.
>
> (2) A thorough and comprehensive experimental study is necessary, including more baselines and more settings.
>
> (3) The paper writing should be improved significantly to be more clear and reader friendly.

---

### Meta-Review · Area_Chair_SGwk · 2024-12-20

**Metareview:**

This paper investigates the generalization capability of symbolic vs black-box optimizers in the context of reinforcement learning. It introduces a symbolic optimizer discovery pipeline based on LLMs and evaluation their generalization against a black-box optimizer OPEN and Adam.

While reviewers agree that topic of developing more generalizable meta-learned optimizer with fewer hyperparameters is well motivated and timely, all of them have concerns on the quality of the proposed approach and the scope of the empirical evaluations. Reviewers maintain their options after the rebuttal, emphasizing the need for boarder evaluations and stronger empirical performance.

**Additional Comments On Reviewer Discussion:**

Authors' rebuttal does not resolve the concerns.

---

### Decision · Program_Chairs · 2025-01-22

Reject